# Immunization as Protection Against Long COVID in the Americas: A Scoping Review

**DOI:** 10.3390/vaccines13080822

**Published:** 2025-07-31

**Authors:** Gabriela Zambrano-Sánchez, Josue Rivadeneira, Carlos Manterola, Tamara Otzen, Luis Fuenmayor-González

**Affiliations:** 1Unidad de Revisiones Sistemáticas y Metaanálisis-URMA, Facultad de Ciencias Médicas, Universidad Central del Ecuador, Quito 170403, Ecuador; dra.gzambrano@gmail.com (G.Z.-S.); j.rivadeneira01@ufromail.cl (J.R.); 2Hospital de Especialidades Eugenio Espejo, Quito 170403, Ecuador; 3Doctorado en Ciencias Médicas, Universidad de La Frontera, Temuco 01145, Chile; carlos.manterola@ufrontera.cl; 4Zero Biomedical Research, Quito 170403, Ecuador

**Keywords:** post-acute COVID-19 syndrome, vaccines, efficacy, Latin America

## Abstract

Introduction: Long COVID syndrome is defined as persistent or new symptoms that appear after an acute SARS-CoV-2 infection and last at least three months without explanation. It is estimated that between 10% and 20% of those infected develop long COVID; however, data is not precise in Latin America. Although high immunization rates have reduced acute symptoms and the pandemic’s impact, there is a lack of evidence of its efficacy in preventing long COVID in the region. Methods: This scoping review followed PRISMA-ScR guidelines. Studies on vaccinated adults with long COVID from Central and South America and the Caribbean were included (Mexico was also considered). A comprehensive search across multiple databases was conducted. Data included study design, participant characteristics, vaccine type, and efficacy outcomes. Results are presented narratively and in tables. Results: Out of 3466 initial records, 8 studies met the inclusion criteria after rigorous selection processes. These studies encompassed populations from Brazil, Mexico, Latin America, and Bonaire, with 11,333 participants, 69.3% of whom were female. Vaccination, particularly with three or more doses, substantially reduces the risk and duration of long COVID. Variability was noted in the definitions and outcomes assessed across studies. Conclusions: This scoping review highlights that SARS-CoV-2 vaccination exhibits potential in reducing the burden of long COVID in the Americas. However, discrepancies in vaccine efficacy were observed depending on the study design, the population studied, and the vaccine regimen employed. Further robust, region-specific investigations are warranted to delineate the effects of vaccination on long COVID outcomes.

## 1. Introduction

As the COVID-19 pandemic continued its course, it became increasingly clear that a significant percentage of those with the disease were experiencing unexplained symptoms after the resolution of the acute illness [1]. A greater consensus was established about these manifestations, and these symptoms were grouped into a syndrome called long COVID. Definitions of long COVID may still vary and continue to be updated [1].

According to the current and most widely adopted definition, these symptoms must be present for at least three months as a continuous, relapsing and remitting, or progressive disease state that affects one or more organ systems [2].

Studies have shown that around 40% of people infected with SARS-CoV-2 can develop symptoms that can be related to long COVID [3,4]. Although the exact number of people living with this condition is uncertain, it is believed that more than 17 million people across the European Region may have experienced it during the first two years of the pandemic [5].

The statistics become less apparent when evaluating the impact of this syndrome in the Americas. However, in several countries, efforts have been made to clarify the situation in the region [6,7]. Despite most symptoms described by those affected by long COVID being mild and self-limiting, there is a considerable proportion of patients who present with long-lasting and debilitating symptoms that lead to disability in otherwise healthy people [8]. Mechanisms recently discovered as possible causes of these disorders have been proposed, such as the persistence of functional viral RNA in various tissues, which can last up to two years [9]. Furthermore, most studies indicate that women are disproportionately affected by this syndrome [8].

Regarding the direct consequences of the virus, the high general rate of immunization with the first vaccine platforms, along with a protective effect due to natural immunity, has significantly limited the acute post-viral effects, also understood as acute post-COVID-19 conditions (PCCs) or post-acute sequelae of COVID-19 (PASCs) [10]. The COVID-19 vaccines were designed to reduce hospitalizations and mortality. Although the use of updated vaccine platforms is based on more updated lineages such as XBB1.5 (which has been prevalent in most of the northern hemisphere), its adoption is still new and in process in Latin America and has even begun to promote the use of mRNA platform for more recent lineages such as JN.1 [11].

Information about the effects of vaccination on long COVID is less clear. For instance, the risk of experiencing an adverse cardiovascular event such as stroke, acute myocardial infarction, venous thromboembolism, or type 2 diabetes increases dramatically after the first year of acute COVID infection [12].

Early vaccination has accelerated recovery from long COVID and has demonstrated efficacy in reducing the risk of developing this condition [13]. This protective effect was observed in people vaccinated with one or two doses, regardless of vaccination status before or after SARS-CoV-2 infection [13]. The emergence of hypertension and diabetes as long COVID conditions underscores the importance of vaccination, particularly among people without a history of vaccination [13].

Regarding the different symptoms presented in long COVID patients, European studies such as COVID Home identified three different phenotypes in convalescent patients: phenotype A, which affects middle-aged patients with few comorbidities and predominantly respiratory symptoms; phenotype B, observed in older women with multimorbidity and characterized by numerous neurological symptoms; and phenotype C, more prevalent in men, similar to phenotype A in the distribution of symptoms but with an average age similar to phenotype B [14].

Other forms of phenotyping, such as those proposed by Zhao et al., further delineate sub-phenotypes based on cardiac-renal, respiratory, musculoskeletal, and digestive manifestations that can also be found in populations in the region and that could be prevented by vaccination [15].

Due to the exploratory nature of our research, we proposed a scoping review against other types of synthesis studies. This scoping review aims to comprehensively respond to the following research question: What was the role of SARS-CoV-2 immunization against the development or severity of long COVID in the Americas region?

## 2. Materials and Methods

### 2.1. Protocol and Registration

This study follows the PRISMA-ScR guidelines for reporting scoping reviews [16]. Before the search started, the protocol for this scoping review was uploaded to the Open Science Framework repository [17]. The protocol was elaborated following the Joanna Briggs Institute (JBI) guidelines for reporting Scoping reviews and Scoping review protocols [18,19].

### 2.2. Eligibility Criteria

This review included observational and experimental studies from Central America, South America, and the Caribbean (Mexico was also considered). The studies involved adults previously vaccinated against SARS-CoV-2 and later developed long COVID, diagnosed clinically by the authors’ criteria. Case reports, case series, opinions, commentaries, systematic reviews, and letters to the editor that did not include original results were excluded. No restrictions on language or year of publication were imposed.

### 2.3. Information Sources

A comprehensive search was performed in the following bibliographic databases: MEDLINE through PubMed, Scopus, Embase, Web of Science, BIREME-BVS, and SciELO. The most recent search was executed on 7 September 2024.

### 2.4. Search

The concepts evaluated were Long COVID and the Effectiveness of vaccination in the prevention of Long COVID. MEsH, DeCS, Emtree, and free terms were used and linked using Boolean operators. We adapted the search strategy for each source of information, which is provided in Appendix A. In addition, the search was supplemented by manual exploration of the bibliographies of all included studies and a search of the gray literature through preprint repositories.

### 2.5. Selection of Sources of Evidence

The authors used the Rayyan Intelligent Systematic Review software (Rayyan Systems Inc., Cambridge, MA, USA) to select evidence. In a two-step process, at least two authors (Z-SG, RJ, and F-GL) independently screened and selected all potentially relevant sources of information.

After eliminating the duplicates, the title and abstract were screened to determine whether the study could respond to the review questions. Afterward, a full-text analysis determined whether they met the selection criteria before inclusion. Disagreements on the inclusion were resolved by consensus.

### 2.6. Data Charting Process

Data extraction was performed using the tools suggested by the JBI guidelines for reporting Scoping reviews [18]. Two reviewers (Z-SG, RJ, and F-GL) did the extraction independently and managed disagreements through consensus. Each reviewer validated the data extraction tool before collecting the data.

### 2.7. Data Items

The variables extracted and assessed were the year and country of publication, study design, participants’ characteristics, number of doses, and the type of vaccine used in the analysis, the long COVID definition, and the efficacy measures reported.

### 2.8. Critical Appraisal of Individual Sources of Evidence

A methodological quality (MQ) assessment was performed independently by two reviewers (RJ and F-GL) using the *Metodología de la Investigación en Cirugía* (MINCir) scale to assess MQ in Therapy Studies [20]. This tool evaluates the MQ in three domains: D1, assessing the rigor of the study design; D2, assessing the sample size and the sample estimation; and D3, which evaluates methodological features such as the objectives, the selection criteria, and the rationale for the sample size and the selection of the design of the study. Studies with a score ≥18 were classified as adequate or high MQ, and those with scores <18 were classified as inadequate or low MQ. Disagreements were managed through consensus.

### 2.9. Synthesis of Results

A qualitative synthesis was performed, and the results are presented narratively and through figures and tables.

## 3. Results

### 3.1. Selection of Evidence Sources

A total of 3466 items were extracted from the sources of information (Figure 1); 1656 items were eliminated due to duplicates, and 1810 articles were evaluated by title and abstract. Twelve articles were analyzed by full text, discarding seven manuscripts. Five studies were included from the sources of information, and three came from the manual search, with eight articles included.

### 3.2. Characteristics and Results of the Sources of Evidence

In total, 62.5% (five) of the studies were published in 2023 [6,7,21,22,23] and 37.5% (three) were published in 2024 [24,25,26]. One article included a population from all of Latin America [6]; four were developed in Brazil [21,22,25,26], two in Mexico [7,24], and one in Bonaire [23]. The most commonly used research design was cohort studies, with 50% (four) (three prospective and one retrospective); 37.5% (three) were cross-sectional studies, and 12.5% (one) were case–control studies (Table 1).

The eight articles analyzed included 11,333 participants, 69.3% (7857) female. Four articles reported the mean and standard deviation of the participants’ age in years; three studies described the median and range, with a minimum of 14 years and a maximum of 89 years; and one manuscript did not describe the age of the research subjects (Table 1).

The majority of the sample consisted of participants from Brazil (8425), followed by Mexico (894), Ecuador (513), and Argentina (480). Panama, Nicaragua, and Costa Rica contributed the fewest participants (Appendix A).

### 3.3. Definitions of Long COVID

The eight articles included define Long COVID, using different characteristics related to the diagnosis of acute COVID-19 infection, the presence of symptoms, and the duration of symptoms compared with the current definition proposed by Ely et al. (Figure 2, Table 2).

NR: Not reported, OR: Odds Ratio, HR: Hazard Ratio;SARS-CoV-2 infection: All eight articles, as a requirement, include a history of acute COVID-19 infection; 37.5% (3) of the studies describe the need for a positive laboratory test for SARS-CoV-2, and 12.5% (1) include suspicion and confirmation of acute infection;Symptoms: The persistence of symptoms from the acute stage was considered by 100% (8) of the studies; remitting and recurrent symptoms and symptom progression were not included in any definition. 62.5% (5) described developing new symptoms after the acute stage of infection;Time of presentation: 100% (8) of the studies describe a specific time of permanence of symptoms following acute infection. 50% (4) consider 12 weeks or more, 12.5% (1) describe 8 weeks or more, and 37.5% (3) use 4 weeks or more as a defining criterion.3.4. Vaccination Status.

Six articles reported the number of doses required to be considered complete vaccination status; 50% (3) used one dose or more of COVID-19 vaccine as a criterion, and the remaining 50% (3) used two doses as a definition.

Two studies included temporality in their definitions of the onset of symptoms of acute SARS-CoV-2 infection. Nuñez et al. [7] required administering a dose of any vaccine at least 14 days before the onset of symptoms, and Berry et al. [23] considered administering a dose 8 weeks after infection.

### 3.4. Reducing the Incidence of Long COVID

Four studies evaluated the risk of developing Long COVID, using different comparisons concerning the number of doses administered and vaccination status (Figure 3).

Administration of one dose: Two studies report a neutral effect on the risk of Long COVID [6,22].Administration of 2 doses: One study described a decrease in the risk of Long COVID [6], and 2 articles reported a neutral effect [21,22].Administration of 3 or more doses: One article reports that the administration of 3 doses has no effect, while four doses decrease the risk of Long COVID [21]. One study describes that after administering three or more doses, the risk decreases [6].Batista et al. evaluated the complete vaccination status, demonstrating that it decreases the risk of Long COVID [25].

### 3.5. The Severity of Symptoms Related to Long COVID

One article studied the change in symptom severity and evaluated 14 symptoms, 13 with neutral results. Describes that vaccination increases the severity of heart palpitations [23].

### 3.6. Duration of Long COVID Symptoms

Two studies considered symptom persistence as an outcome. Fuller et al. [26] found that vaccination with two or more doses decreases Long COVID symptoms. Del Carpio-Orantes et al. [24] did not identify a significant decrease in vaccinated participants’ neurologic, cardiac, pulmonary, gastrointestinal, and musculoskeletal symptoms. One study demonstrated that administering one or more doses before acute COVID-19 infection decreases the time to resolution of long COVID [7].

## 4. Discussion

Post-COVID-19 conditions (PCCs) affect approximately 20–30% of unvaccinated individuals between three and six months following SARS-CoV-2 infection [3]. This highlights the importance of vaccination coverage in reducing the long-term impact of COVID-19. Latin America experienced unequal coverage of the COVID-19 primary vaccination series (first and second doses), despite surpassing a 70% immunization rate for most of the population by April 2022 [1].

Although efforts in Latin America to assess the severity of Long COVID have been limited, they are notable, such as the Chilean COVID-19 Biorepository. In this project, participants were categorized into six severity groups: asymptomatic (*n* = 169); mild cases (*n* = 1712); hospitalized without oxygen support (*n* = 146); hospitalized with oxygen support (*n* = 71); and critically ill patients (*n* = 151) requiring advanced respiratory support such as mechanical ventilation, continuous positive airway pressure (CPAP), bilevel positive airway pressure (BiPAP), or high-flow oxygen therapies like Optiflow. They also analyzed the participants’ genetic ancestry. The results showed that 13.4% (SD 17.2%) had a northern component linked to Aymara and Quechua populations, while 30.6% (SD 15.0%) had a southern component associated with Mapuche ancestry [27]. Initiatives like this could help improve the understanding of this condition in the region.

Given this context, it is essential to contextualize long COVID as an under-researched syndrome among Latin American patients. The limited number of studies in the region addressing vaccine-based prevention of PCCs underscores this research gap. Most meta-analyses have been conducted in North American and European populations [28,29], excluding regional cohorts due to the lack of systematic patient follow-up protocols in Latin American countries and the underutilization of the ICD-10 diagnostic code U07.9 for proper case registration [30]. Moreover, the challenges of public healthcare systems and the limited capacity of healthcare providers to recognize the broad phenotypic spectrum of this clinical entity complicate its identification.

Studies characterizing the long COVID syndrome have predominantly been conducted in high-income countries (HICs). However, the burden of long COVID in low- and middle-income countries (LMICs) remains insufficiently explored [31].

This scoping review underscores the complex relationship between SARS-CoV-2 vaccination and long COVID in the Americas. Despite considerable advances in understanding long COVID, including its clinical definitions and phenotypes, the role of immunization in mitigating the syndrome remains unclear in the region.

### 4.1. What Is Already Known About This Topic

The evidence suggests that SARS-CoV-2 immunization, especially with multiple doses, could provide protection against long COVID. For instance, in a systematic review and meta-analysis, Watanabe et al. reported that two vaccine doses were related to a lower risk of developing long COVID compared to no vaccination (OR = 0.64; 95% CI 0.45–0.92) and a significant effect compared to one-dose vaccination (OR = 0.60; 95% CI, 0.43–0.83) [28].

### 4.2. Main Findings

In Latin America, some studies found no effect with one or two doses of vaccination [21,22], while others found a risk reduction with three or more doses [6,22]. These results contrast with large population-based European studies where COVID-19 vaccination was strongly associated with decreased probability of developing long COVID (HR = 0.48; 95% CI 0.34–0.68) [32]. The findings emphasize the necessity for consistency in defining the “complete vaccination” status in Latin American studies and its temporal association with SARS-CoV-2 infection.

The protective effect reported with greater vaccine doses is consistent with previously proposed mechanisms, such as reduced virus persistence, microvascular and endothelial dysfunction, dysfunctional vagal signaling, and immune response dysregulation [9,11]. However, the neutral effects found in certain studies require additional research into individual and geographical characteristics that may affect vaccine efficacy, such as sex [33], age [13,32], comorbidities [34], time of vaccination (before or after SARS-CoV-2 infection [13,28] or during delta or omicron phases [35]) and type of vaccines [32].

In Mexican and Brazilian cohorts, PCC patients were older, with more comorbidities and a higher proportion of unvaccinated individuals requiring supplemental oxygen during the acute phase of infection [7,22].

While Marra and Núñez’s cohorts identified men as more affected [7,22], Angarita’s Hispanic cohort showed, via adjusted logistic regression models, higher odds of PCC among women and unvaccinated individuals with hypertension or diabetes [6], consistent with other Mexican series [24].

Importantly, this multicenter cohort, which also included Ecuadorian participants [6], described a higher prevalence of sequelae in unvaccinated men or those with only one dose. These individuals reported higher rates of respiratory and metabolic disorders and required supplemental oxygen during acute infection.

The study by Marra et al. details vaccine-based prevention of SARS-CoV-2 sequelae [22]. This case–control study, conducted among healthcare workers in São Paulo between 2020 and 2022, reported an accumulated incidence within a 95% confidence interval for this cohort. Like other Latin American studies [6,7,22], the most prevalent comorbidities included hypertension and diabetes mellitus, even among healthcare personnel. Notably, this study is among the few that incorporated genomic sequencing to identify circulating variants. During the study period, most samples corresponded to the Delta (δ) variant, followed by Omicron and Gamma [22]. In this cohort, most vaccinated individuals had received at least one dose and were healthcare workers, often immunized with heterologous regimens. Vaccinated cases were less likely to develop PCCs than those infected before immunization. Reinfection with SARS-CoV-2 was identified as a major risk factor for long COVID, while continued immunization was associated with reduced risk [19].

The effect of vaccination on the severity and duration of long COVID is less clear because of the lack of appropriate definitions. Large studies demonstrated that vaccination reduced the incidence of severe thromboembolic and cardiovascular complications of long COVID [36]. Locally, Brazil stands out as the Latin American country with the most research initiatives in this field, The one by Fuller et al. at the Oswaldo Cruz Foundation in Rio de Janeiro recruited a cohort of 276 adults to compare neutralizing antibody levels across three groups: individuals who remained asymptomatic, those who experienced symptoms for up to three months, and those with symptoms persisting for nine months or more [26]. This study demonstrated that individuals with prolonged symptoms had significantly lower anti-S1 IgG levels than asymptomatic controls, with clinical manifestations such as chronic fatigue. In contrast, post-acute asymptomatic individuals displayed high neutralizing antibody levels. The prevalence of persistent symptoms decreased over time, with only 5% reporting symptoms one year post-infection. Notably, individuals who had received multiple booster doses had a significantly reduced risk of developing PCCs [26]. It was found that immunization with two or more doses reduced the persistence of symptoms, confirming its possible effect in speeding recovery, as reported by Peluso et al. [9].

In line with previous studies, another patient cohort was conducted in Complexo da Maré, a network of 16 slums in Rio de Janeiro, Brazil. The study aimed to estimate the prevalence of long COVID, assess its impact on quality of life and functional status, and identify risk factors associated with worse outcomes. The primary outcome was self-perceived recovery ≥90 days after symptom onset. At follow-up, 20% of participants (143/706) reported not feeling fully recovered. Non-recovery was associated with older age, female sex, pre-existing comorbidities (especially hypertension, diabetes, and obesity), lower education, hospitalization, and dyspnea during acute illness. Secondary outcomes involved declines in health-related quality of life, functional limitations, and persistent symptoms. Compared to the pre-COVID period, 32% of participants reported health deterioration. Increases were observed in anxiety/depression (from 23% to 32%), mobility issues (7% to 16%), pain/discomfort (14% to 25%), and limitations in daily activities (3% to 7%). Female sex, comorbidities, and dyspnea during the acute phase were independently associated with worse outcomes, including persistent dyspnea and decreased quality of life [31].

The Mexican hospitalized cohort study is one of the few to offer phenotypic categorization of PCC s. With a median follow-up of 405 days, six phenotypes were described: respiratory, mucocutaneous, neurological impairment, functional, gastrointestinal, and mood/sleep/cognitive disorders, encompassing a total of 23 symptoms [24]. They reported no significant reduction in specific symptoms among vaccinated individuals. These contradictory results can be related to different long COVID phenotypes and their response to immunization, as previously shown in other cohorts [35].

When comparing this study to the cohort by Núñez et al. in Mexico [7], which described the frequency, predictors, and duration of 23 PCC-related symptoms, hypertension and diabetes again emerged as factors associated with prolonged illness. Similarly, the cohort by Carpio-Orantes, which identified 138 adults with disease burden linked to the triad of diabetes mellitus, hypertension, and obesity (also reported in the cohorts by Núñez and Arango), did not find a significant association between vaccination history and PCCs, a notable and unique finding. This study also reported a high incidence of autoimmune disorders, particularly among women, patterns consistent with international reports but less frequently observed in Latin American cohorts [24].

Notably, prior predictive modeling conducted in Mexican cohorts has identified myalgia, tachycardia, and antibiotic use as significant risk factors for the development of long COVID. In contrast, higher educational attainment and blood type B appear to confer a protective effect [37].

Additionally, assessing adverse effects was another relevant issue in the selected studies. Berry et al. [23] reported an increase in the severity of heart palpitations following immunization in select populations, emphasizing the need to weigh the benefits of vaccination against the potential hazards described in numerous studies [38,39,40].

### 4.3. Implications for Public Health in the Americas. A Call to Action

The vast majority of studies evaluating the impact of immunization in long COVID have been developed in regions different than Latin America. Moreover, differences in regional vaccine access and adoption underline critical public health challenges in these countries. With most included studies originating from Brazil and Mexico, the findings may not fully represent the diverse contexts of the region. Expanding research efforts to include underrepresented countries, such as those in Central America, the Caribbean, and the Andean region, is essential for a comprehensive understanding of the impact of vaccination on long COVID.

The emergence of new vaccine platforms designed to cover new lineages offers promising alternatives for reducing long COVID risks. However, their implementation in the Americas has been slow, highlighting the need for robust vaccination campaigns and policies to ensure equitable access.

### 4.4. Limitations

This review is subject to limitations related to small sample sizes, varied methodologies, and inconsistent definitions of long COVID. Furthermore, the absence of data on specific populations and the restricted scope of observational designs limit the generalizability of findings. Future regional research should prioritize longitudinal and interventional studies to determine the causal association between vaccination and long-term COVID outcomes.

## 5. Conclusions

In Latin America, vaccination against SARS-CoV-2 appears to reduce the incidence and duration of long COVID, but its effectiveness varies depending on the population and vaccine regimen. These findings highlight the necessity of ongoing research and public health initiatives to improve immunization strategies, especially in areas with varying socioeconomic and healthcare landscapes.

## Figures and Tables

**Figure 1 vaccines-13-00822-f001:**
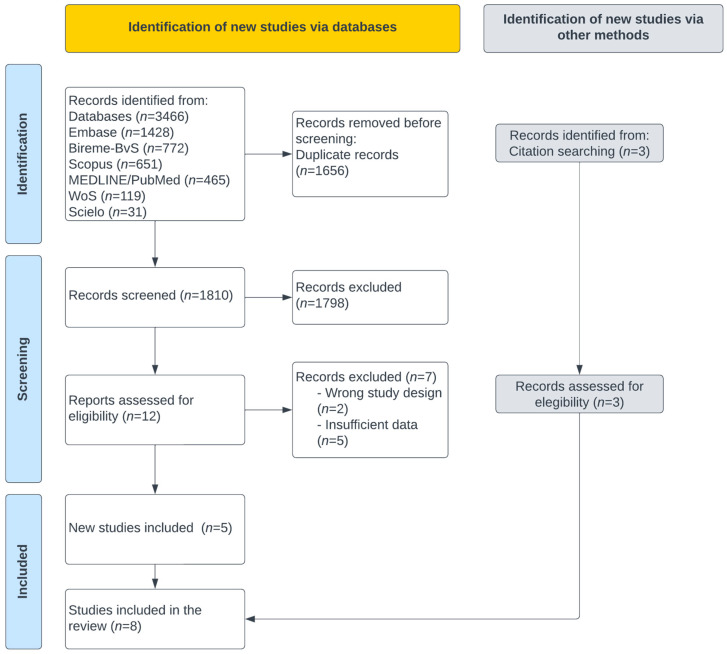
PRISMA flow diagram for studies selection.

**Figure 2 vaccines-13-00822-f002:**
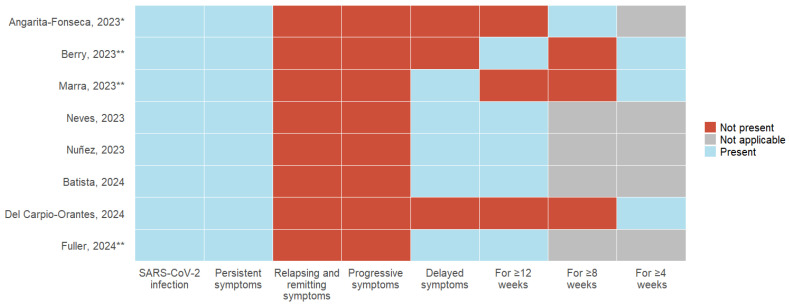
Variability in long COVID definitions: Comparison of diagnostic criteria, symptoms, and duration across primary articles [6,7,21,22,23,24,25,26]. * *History of probable or confirmed SARS-CoV-2 infection*; ** *SARS-CoV-2 infection confirmed with test*.

**Figure 3 vaccines-13-00822-f003:**
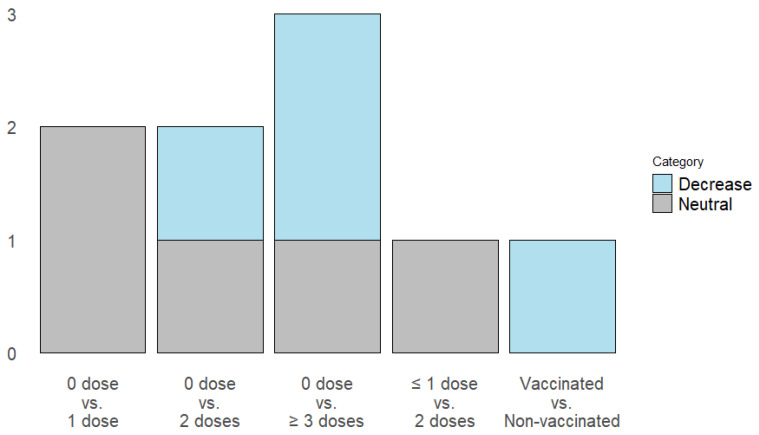
Impact of vaccination on long COVID incidence: Dose-dependent risk reduction of primary outcomes.

**Table 1 vaccines-13-00822-t001:** Characteristics of the included studies.

Author, Year	Country	Design	Number of Participants		Methodological Quality
			Sex *n*, (%)	Age, Years	
Angarita-Fonseca, 2023 [6]	Latin-America	Cross-sectional study	Men: 840 (34.1); Women:1626 (65.9)	Mean (SD): 39.5 (53.3)	Inadequate or Low
Berry, 2023 [23]	Bonaire	Retrospective cohort study	Men: 10 (21.2); Women: 37 (78.8)	Median (range): 47 (14–89)	Adequate or High
Marra, 2023 [22]	Brazil	Case–control study	Men: 1950 (27.6); Women: 5101 (72.4)	Mean (SD): General: 37.5 (NR) Cases: 38.1 (8.7); Controls: 37.2 (9.0)	Inadequate or Low
Neves, 2023 [21]	Brazil	Prospective cohort study	Men: 338 (56.1); Women: 264 (43.9)	Mean (SD): 51 (12)	Adequate or High
Nuñez, 2023 [7]	Mexico	Prospective cohort study	Men: 126 (65.6); Women: 66 (34.4)	Median (range): 53 (45–64)	Inadequate or Low
Batista, 2024 [25]	Brazil	Cross-sectional study	Men: 59 (11.9); Women: 437 (88.1)	NR	Inadequate or Low
Del Carpio-Orantes, 2024 [24]	Mexico	Cross-sectional study	Men: 65 (32,0); Women: 138 (68,0%)	Mean (SD): 41.8 (11.3)	Inadequate or Low
Fuller, 2024 [26]	Brazil	Prospective cohort study	Men: 88 (31.8); Women: 188 (68.2)	Median (range): 45 (18–88)	Adequate or High

NR: Not reported.

**Table 2 vaccines-13-00822-t002:** Efficacy of immunization against long COVID.

Author, Year	“Fully Vaccinated” Status	Vaccine Type	Long COVID Definition	Efficacy Measures
Angarita-Fonseca, 2023 [6]	Two doses	NR	Individuals with a history of probable or confirmed SARS-CoV-2 infection, usually 3 months from the onset of COVID-19 with symptoms that last for at least 2 months and cannot be explained by an alternative diagnosis.	Outcome: Risk of development of long COVID. Multivariable logistic regression. 1 dose: OR: 0.82; 95% CI 0.6–1.1 2 doses: OR: 0.75; 95% CI 0.6–0.9 3 or more doses: OR: 0.81; 95% CI 0.6–1.0.
Berry, 2023 [23]	At least one dose of the Pfizer vaccine at least 8 weeks after SARS-CoV-2 infection	mRNA: 36, Unvaccinated: 11	Individuals with a laboratory-confirmed SARS-CoV-2 positive test result, for whom at least one symptom self-attributed to the experienced SARS-CoV-2 infection lasted longer than four weeks.	Outcome: Self-reported change in symptom severity. Multiple covariate adjusted linear regression model. Regression coefficients and 95% CI: Cough: −0.36; 95% CI −1.11–0.39; Heart palpitations: 0.60; 95% CI 0.18–1.02
Marra, 2023 [22]	Analysis were performed whether 1, 2, 3, or 4 doses were administered.	Inactivated virus= 3259; Viral vector= 3255; mRNA = 148	Signs and symptoms that developed during or following a SARS-CoV-2 RT-PCR confirmed infection, continued for >4 weeks, and could not explained by an alternative diagnosis.	Outcome: Risk of development of long COVID. Logistic Regression multivariable analysis. 0 vs. 1 doses: OR:0.91; 95%CI: 0.60–1.39; 0 vs. 2 doses: OR: 1.17; 95%CI: 0.79–1.76; 0 vs. 3 doses: OR: 0.63; 95%CI: 0.39–1.02; 0 vs. 4 doses: OR: 0.05; 95%CI: 0.01–0.19.
Neves, 2023 [21]	Two doses	Homologous regimens: Inactivated Whole-virion vaccine: 189; mRNA vaccine: 24; Viral-vector vaccine:96	Physical complaints newly developed during or after the acute phase, persisting for >12 weeks, and not explained by an alternative diagnosis.	Outcome: risk of long COVID. HR: 0.89; 95% CI: 0.57–1.41
Nuñez, 2023 [7]	At least one dose of any SARS-CoV-2 vaccine at least 14 days before the date on which symptoms of acute infection began	NR	Patients experiencing any symptoms not present before acute COVID-19 onset, and that persisted for longer than 90 days after acute COVID-19 onset.	Outcome: probability to experience a shorter time to long COVID resolution. HR: 3.16; 95%CI 1.21–8.26
Batista, 2024 [25]	NR	NR	Symptoms that remain or appear for the first time within three months of SARS-CoV-2 infection.	NR
Del Carpio-Orantes, 2024 [24]	One dose or more	NR	Persistence of COVID-19 symptoms four weeks after the acute episode.	Outcome: probability to experience: Neurological symptoms: OR: 3.768 (CI 0.684–20.766); Cardiac symptoms: OR: 0.213 (CI 0.028–1.640); Pulmonary symptoms: OR: 1.649 (CI 0.645–1.640); Gastrointestinal symptoms: OR: 0.391 (CI 0.087–1.753); Musculoskeletal symptoms: OR: 0.422 (CI 0.138–1.286)
Fuller, 2024 [26]	Two or more doses	NR	Symptoms that began within three months of the positive SARS-CoV-2 test.	Outcome: Persistence of Long COVID in not fully vaccinated people. HR: 1·96, 95 % CI: 1·03–3·7

## Data Availability

Search results are available under reasonable request.

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
