# Peer review of "Immunization as Protection Against Long COVID in the Americas: A Scoping Review"

_vaccines, 2025, doi:10.3390/vaccines13080822_

Round 1

Reviewer 1 Report

Comments and Suggestions for Authors

The manuscript submitted to Vaccines by Tamara Otzen and coauthors looks like a hastily made analysis of several references. In terms of the level of presentation of the material and the depth of analysis of the sources, the article does not correspond to the level of Vaccines.

In addition, in this article, diagrams and tables are of primary importance, their level of scholar value is extremely low. Here are some specific comments, the correction of which, however, will not make the article suitable for publication in Vaccines

1) Table 1 should be moved to Supplementary

2) The information presented in Table 2 is extremely difficult for the reader to understand. The material is presented in the form of text, instead of being distributed over a larger number of cells (columns, rows). The value of this information for the reader is questionable; it is easier to read the articles themselves than to understand a hastily assembled table.

3) The letters in the lower diagonal legend to Figures 2 and 3 are stuck together and are difficult to read

4) In Fig. 3 red color is not presented

5) The Conclusion section contains trivial conclusions that could have been formulated without conducting research and reading 35 references. Could the authors really only extract this information from them? Where are the numerical values, correlations, statistics, differences between the groups?

Comments on the Quality of English Language

The level on English is not the major issue, but the manuscript is recommended to be reviewed by professional scholar editor.

Author Response

Response to the reviewers. Summary

Thank you very much for taking the time to review this manuscript. Please find the detailed responses below and the corresponding corrections highlighted in the re-submitted file.

Point-by-Point Response to Comments and Suggestions for Authors

Assitant Editor:

  1. If possible, could you please provide us with Dr. Gabriela Zambrano-Sánchez’s institutional email address (e.g., @xxx.edu.cn)?

Response from the authors:

Dr. Zambrano-Sánchez’s institutional email adress is the following: patricia.zambrano@hee.gob.ec

Reviewer 1:

  • The manuscript submitted to Vaccines by Tamara Otzen and coauthors looks like a hastily made analysis of several references. In terms of the level of presentation of the material and the depth of analysis of the sources, the article does not correspond to the level of Vaccines. In addition, in this article, diagrams and tables are of primary importance, their level of scholar value is extremely low. Here are some specific comments, the correction of which, however, will not make the article suitable for publication in Vaccines
  • Table 1 should be moved to Supplementary

Response from the authors:

Thank you for pointing this out. The first version of the manuscript has been restructured to maintain rigor enough for publication in Vaccines. On the other hand, we believe that the main results of a Review are presented through tables, especially in circumstances where a quantitative data synthesis is not possible. For this reason, we kept Table 1 in its current form but have improved Table 2 to present the information in a simpler form.

  • The information presented in Table 2 is extremely difficult for the reader to understand. The material is presented in the form of text, instead of being distributed over a larger number of cells (columns, rows). The value of this information for the reader is questionable; it is easier to read the articles themselves than to understand a hastily assembled table.

Response from the authors:

We apologize for the difficulties reading this Table. However, we have to argue that the platform used to upload the manuscript did not allow us to upload this table in a landscape orientation, which could make the reading much easier.

  • The letters in the lower diagonal legend to Figures 2 and 3 are stuck together and are difficult to read

Response from the authors:

We sincerely apologize for this issue. The legends have been fixed.

  • In Fig. 3 red color is not present

Response from the authors:

Thank you for your feedback. Our objective was to demonstrate the lack of studies indicating an increase in Long COVID incidence. We have retired the red color in Figure 3.

  • The Conclusion section contains trivial conclusions that could have been formulated without conducting research and reading 35 references. Could the authors really only extract this information from them? Where are the numerical values, correlations, statistics, differences between the groups?

Response from the authors:

We agree with your insights. However, we believe the reviewer refers to the Discussion section, where numerical values are normally explained. We have restructured this section, improving the findings discussion and comparisons among studies.

Reviewer 2 Report

Comments and Suggestions for Authors

This review entitled “Immunization as protection against long COVID in the Americas: A scoping review” is timely and relevant, addressing a pressing public health issue: the role of SARS-CoV-2 immunization in mitigating long COVID in Latin America. It is well-structured and adheres to PRISMA-ScR guidelines, offering a comprehensive synthesis of available data. The focus on long COVID and vaccination efficacy is crucial as global healthcare systems address long-term pandemic consequences. The regional focus on Latin America fills a significant gap in the literature. Follows established scoping review methodology (PRISMA-ScR and JBI). Use of multiple databases and gray literature improves comprehensiveness. The manuscript clearly defines eligibility criteria, geographic scope, and study types included. The authors critically evaluate inconsistent findings and provide plausible explanations for differences across studies. They appropriately highlight variability in definitions of long COVID and vaccine regimens. The authors stress the need for equitable vaccine access and further research in underrepresented areas, which enhances policy relevance.

However, these aspects should be improved:

  1. Inconsistent use of terms such as “long COVID,” “post-COVID condition (PCC),” and “long-term COVID outcomes.” Please standardize terminology throughout and align definitions more clearly with WHO and CDC standards.
  2. No quality appraisal was performed. While this is not mandatory for scoping reviews, brief commentary on study design strength/limitations would add interpretive value. Please include a concise evaluation of the methodological quality or risk of bias to guide readers on evidence reliability.
  3. Tables (especially Table 2) are dense and difficult to follow due to inconsistent formatting. Please improve layout by using concise column headers, use color coding or grouping for vaccine types and dose levels to enhance readability, add footnotes where needed to clarify abbreviations or assumptions.
  4. Though synthesis is primarily qualitative, pooled data (e.g., participant demographics, proportion experiencing long COVID) could be numerically summarized. Please consider adding a summary table aggregating key metrics across studies (e.g., mean effect size, median follow-up duration, % with symptoms at 12 weeks).
  5. The role of comorbidities, prior infection severity, and vaccine type is discussed but not explored in depth. Please expand discussion on how these factors might influence observed heterogeneity in vaccine efficacy.
  6. There are some minor Language and Stylistic Issues, for example, "symptomatology"- better phrased as "symptom profile" or "symptoms." “NR” (not reported) in tables is overused without context. Please perform a thorough language polish for clarity and conciseness.
Comments on the Quality of English Language

This review entitled “Immunization as protection against long COVID in the Americas: A scoping review” is timely and relevant, addressing a pressing public health issue: the role of SARS-CoV-2 immunization in mitigating long COVID in Latin America. It is well-structured and adheres to PRISMA-ScR guidelines, offering a comprehensive synthesis of available data. The focus on long COVID and vaccination efficacy is crucial as global healthcare systems address long-term pandemic consequences. The regional focus on Latin America fills a significant gap in the literature. Follows established scoping review methodology (PRISMA-ScR and JBI). Use of multiple databases and gray literature improves comprehensiveness. The manuscript clearly defines eligibility criteria, geographic scope, and study types included. The authors critically evaluate inconsistent findings and provide plausible explanations for differences across studies. They appropriately highlight variability in definitions of long COVID and vaccine regimens. The authors stress the need for equitable vaccine access and further research in underrepresented areas, which enhances policy relevance.

However, these aspects should be improved:

  1. Inconsistent use of terms such as “long COVID,” “post-COVID condition (PCC),” and “long-term COVID outcomes.” Please standardize terminology throughout and align definitions more clearly with WHO and CDC standards.
  2. No quality appraisal was performed. While this is not mandatory for scoping reviews, brief commentary on study design strength/limitations would add interpretive value. Please include a concise evaluation of the methodological quality or risk of bias to guide readers on evidence reliability.
  3. Tables (especially Table 2) are dense and difficult to follow due to inconsistent formatting. Please improve layout by using concise column headers, use color coding or grouping for vaccine types and dose levels to enhance readability, add footnotes where needed to clarify abbreviations or assumptions.
  4. Though synthesis is primarily qualitative, pooled data (e.g., participant demographics, proportion experiencing long COVID) could be numerically summarized. Please consider adding a summary table aggregating key metrics across studies (e.g., mean effect size, median follow-up duration, % with symptoms at 12 weeks).
  5. The role of comorbidities, prior infection severity, and vaccine type is discussed but not explored in depth. Please expand discussion on how these factors might influence observed heterogeneity in vaccine efficacy.
  6. There are some minor Language and Stylistic Issues, for example, "symptomatology"- better phrased as "symptom profile" or "symptoms." “NR” (not reported) in tables is overused without context. Please perform a thorough language polish for clarity and conciseness.

Author Response

This review entitled “Immunization as protection against long COVID in the Americas: A scoping review” is timely and relevant, addressing a pressing public health issue: the role of SARS-CoV-2 immunization in mitigating long COVID in Latin America. It is well-structured and adheres to PRISMA-ScR guidelines, offering a comprehensive synthesis of available data. The focus on long COVID and vaccination efficacy is crucial as global healthcare systems address long-term pandemic consequences. The regional focus on Latin America fills a significant gap in the literature. Follows established scoping review methodology (PRISMA-ScR and JBI). Use of multiple databases and gray literature improves comprehensiveness. The manuscript clearly defines eligibility criteria, geographic scope, and study types included. The authors critically evaluate inconsistent findings and provide plausible explanations for differences across studies. They appropriately highlight variability in definitions of long COVID and vaccine regimens. The authors stress the need for equitable vaccine access and further research in underrepresented areas, which enhances policy relevance.

However, these aspects should be improved:

  • Inconsistent use of terms such as “long COVID,” “post-COVID condition (PCC),” and “long-term COVID outcomes.” Please standardize terminology throughout and align definitions more clearly with WHO and CDC standards.

Response from the authors:

We appreciate your commentary. In the whole manuscript, the term “long COVID” has been used except for very few situations where the term “PCC” facilitated the text reading.

  • No quality appraisal was performed. While this is not mandatory for scoping reviews, brief commentary on study design strength/limitations would add interpretive value. Please include a concise evaluation of the methodological quality or risk of bias to guide readers on evidence reliability.

Response from the authors:

Thank you for pointing this out. We have performed a methodological quality assessment by using the MinCir in Therapy Studies tool.

  • Tables (especially Table 2) are dense and difficult to follow due to inconsistent formatting. Please improve the layout by using concise column headers, using color coding or grouping for vaccine types and dose levels to enhance readability, and adding footnotes where needed to clarify abbreviations or assumptions.

Response from the authors:

We agree with the reviewer’s insight. Table 2 has been restructured to improve its clarity

  • Though synthesis is primarily qualitative, pooled data (e.g., participant demographics, proportion experiencing long COVID) could be numerically summarized. Please consider adding a summary table aggregating key metrics across studies (e.g., mean effect size, median follow-up duration, % with symptoms at 12 weeks).

Response from the authors:

Thank you for this valuable recommendation. We agree with the reviewer’s viewpoint. Numerical relevant data is now better displayed in Tables 1 and 2, including key metrics and outcome results.

  • The role of comorbidities, prior infection severity, and vaccine type is discussed but not explored in depth. Please expand the discussion on how these factors might influence observed heterogeneity in vaccine efficacy.

Response from the authors:

We agree with the reviewer’s suggestion. These important variables are further assessed in the newly structured discussion section.

  • There are some minor Language and Stylistic Issues, for example, "symptomatology"- better phrased as "symptom profile" or "symptoms." “NR” (not reported) in tables is overused without context. Please perform a thorough language polish for clarity and conciseness.

Response from the authors:

We sincerely appreciate the Reviewer’s constructive suggestions. We strongly believe they will improve the manuscript’s overall quality and rigor. We have thoroughly checked for misspellings and word misuse. Besides that, a professional English review will be performed prior to the new review.

Reviewer 3 Report

Comments and Suggestions for Authors

This manuscript summarizes the evidence on the role of vaccines in the development of long-term COVID in Latin America. However, its approach is so broad and superficial that it leaves the impression that it fails to provide any clear original concepts.
The abstract does not provide any specific results. Please add any relevant data (statistical value) that demonstrates the results found and supports the conclusion.

I believe that, although the manuscript is suitable for a scoping review, this review does not provide anything potentially relevant or any specific data, either as a researcher in the field or as a physician to guide my decision-making. I consider the result to be very ambiguous and relies on the observations and conclusions already reported in the included studies, without providing a comprehensive view of all patients. I suggest a meta-analysis should be conducted, including subjects whose development of long-term COVID is assessed and their vaccination status is mentioned. This could include both vaccinated and unvaccinated patients (studies could include patients in which none were vaccinated, others that included only vaccinated patients, and studies that included both unvaccinated and vaccinated patients). This could increase the sample size and could provide a value that reflects the role of vaccines. The issue of variation in efficacy across vaccine types and doses is also relevant, although a stratified analysis would have to be evaluated depending on the number and types of studies included.

Author Response

  1. This manuscript summarizes the evidence on the role of vaccines in the development of long-term COVID in Latin America. However, its approach is so broad and superficial that it leaves the impression that it fails to provide any clear original concepts.
    The abstract does not provide any specific results. Please add any relevant data (statistical value) that demonstrates the results found and supports the conclusion.

Response from the authors:

We appreciate the reviewer’s viewpoint. However, we believe that, considering the exploratory nature of Scoping reviews, and by the fact that no new data results from this review, the abstract should not contain any numerical data obtained from primary studies. Instead, the characteristics of the selected studies should be placed.

  1. I believe that, although the manuscript is suitable for a scoping review, this review does not provide anything potentially relevant or any specific data, either as a researcher in the field or as a physician to guide my decision-making. I consider the result to be very ambiguous and relies on the observations and conclusions already reported in the included studies, without providing a comprehensive view of all patients.

Response from the authors:

We appreciate the reviewer’s feedback. We partially agree with this statement. The objective of this manuscript was to assess the role of SARS-CoV-2 immunization against the development or severity of long COVID in the Americas region. For this purpose, we compared the suitability of a Scoping Review or other types of Synthesis methods, such as a Systematic Review with or without a meta-analysis. Due to the exploratory nature of this Review, and the apparent lack of appropriately designed studies for a most rigorous synthesis during preliminary searches, we consider that a Scoping review could be the most accurate design.

  1. I suggest a meta-analysis should be conducted, including subjects whose development of long COVID is assessed and their vaccination status is mentioned. This could include both vaccinated and unvaccinated patients (studies could include patients in which none were vaccinated, others that included only vaccinated patients, and studies that included both unvaccinated and vaccinated patients). This could increase the sample size and could provide a value that reflects the role of vaccines. The issue of variation in efficacy across vaccine types and doses is also relevant, although a stratified analysis would have to be evaluated depending on the number and types of studies included.

Response from the authors:

We agree with the reviewer’s suggestion. A Systematic review with meta-analysis would definitely improve the quality of this manuscript. However, because of the lack of randomized clinical trials, the high heterogeneity among studies’ designs, and the inadequate or low methodological quality of the majority of studies, we consider that performing a Scoping Review was the best option; and meta-analyses are not feasible in this type of studies.

Round 2

Reviewer 1 Report

Comments and Suggestions for Authors

The authors made a minor changes in the manuscript. The reviewer suggests, that the current state, the level of presentation of the material and the depth of analysis of the sources, is not corresponding to the scientific level of Vaccines.

Comments on the Quality of English Language

The level on English is not the major issue, the text currently still lacks clarity of presentation, which could be improved by a professional scientific English editor

Author Response

Response from the authors:

Thank you for your feedback. We have tried to get the highest methodological standards that a Scoping review could achieve. However, we appreciate the time spent reviewing the manuscript.

Reviewer 3 Report

Comments and Suggestions for Authors

I understand that the design is well implemented and consistent with the exploratory nature of Scoping Review, and although the contribution is limited, the approach is valuable. However, I believe it would be appropriate to briefly present an overview of long COVID in Latin America, which would surely give the reader a broader view. I have seen a paragraph added/modified in the discussions on this point (lines 317-325, “When comparing this study to the cohort by Núñez et al. in Mexico[7], which described 317 the frequency, predictors….”). In this paragraph or another, I suggest mentioning other studies investigating long COVID in Latin America, regardless of whether they compare vaccinated vs. unvaccinated patients, such as:
https://pubmed.ncbi.nlm.nih.gov/39596552/
https://pubmed.ncbi.nlm.nih.gov/36673565/
https://pubmed.ncbi.nlm.nih.gov/39100241/
among others.

Author Response

Response from the authors:

We strongly agree and sincerely appreciate the reviewer’s valuable commentary. Studies that do not assess vaccinated vs. unvaccinated patients have relevant information that we omitted in previous versions of the manuscript.

We have deepened the discussion section, highlighting important data from studies assessing Long COVID in Latin America, as suggested by the reviewer.

Round 3

Reviewer 3 Report

Comments and Suggestions for Authors

The manuscript may be accepted and published

Author Response

Dear Reviewer

We have formatted Table 2 into portrait orientation as requested by the Editor.

Thank you very much for your insights and support.